# Hemostasis System and Plasminogen Activity in Retrochorial Hematoma in the First Trimester of Pregnancy

**DOI:** 10.3390/biomedicines10092284

**Published:** 2022-09-14

**Authors:** Natalia B. Tikhonova, Natalya B. Kuznetsova, Andrey P. Milovanov, Eugene I. Goufman, Tatiana V. Fokina, Andrey P. Aleksankin, Valentina V. Aleksankina, Irina I. Stepanova, Alexandr A. Stepanov, Marina N. Boltovskaya, Natalia V. Nizyaeva

**Affiliations:** 1Avtsyn Research Institute of Human Morphology of FSBI “Petrovsky National Research Centre of Surgery”, 117418 Moscow, Russia; 2Center for Stimulation Training, Rostov State Medical University, Ministry of Health of Russia, 344022 Rostov-on-Don, Russia

**Keywords:** plasminogen, plasminogen activity, autoantibodies to plasminogen, endometrium, retrochorial hematoma, first trimester of pregnancy

## Abstract

(1) Background: The components of the fibrinolytic system and its main component, plasminogen, play a key role in the first months of pregnancy. The effect of autoantibodies interacting with plasminogen in the formation of retrochorial hematoma is unknown. The aim of our study was to determine the role of plasminogen and IgA, IgM, and IgG, which bind to plasminogen, in retrochorial hematoma. (2) Methods: Prothrombin time (PT), thrombin time (TT), partial activated thromboplastin time (aPTT), soluble fibrin-monomer complex (SFMC), D-dimer, plasminogen activity (%Plg), plasminogen concentration (Plg), and the levels of IgG (IgG-Plg), IgM (IgM-Plg), IgA (IgA-Plg) interacting with plasminogen were determined in plasma samples of 57 women with normal pregnancy and 16 with retrochorial hematoma. (3) Results: %Plg in plasma samples from women with retrochorial hematoma was significantly lower than in plasma samples from women with normal pregnancy. The diagnostic significance of %Plg in the ROC analysis was AUC = 0.85. A direct correlation was found between aPTT and the level of autologous IgM interacting with plasminogen. (4) Conclusions: A decrease in the activity of plasminogen in the blood serum of women in the first trimester of pregnancy may indicate disturbances in the hemostasis system and the formation of retrochorial hematoma. According to the results of the study, it is possible to recommend the determination of plasminogen activity in the management of pregnant women in gynecological practice.

## 1. Introduction

According to data in the literature, reproductive losses in the first trimester account for 12–15% [1] of all pregnancies and 70% of all miscarriages [2]. Besides spontaneous abortions without delay evacuation of the product conceiving, obstetricians distinguish missed pregnancy in case of being the product of conceiving in the uterine cavity. The study of hemostasis is known to be an important part of monitoring the course of pregnancy. Hemostasis system changes in physiological pregnancy are an adaptive process [3] necessary for the normal functioning of the feto-placental complex and limiting blood loss during childbirth.

During gestation in the mother’s body, there is an accumulation of prothrombotic reserve [4], an increase in the level of fibrinogen [5], and an increase in the activity of factors of the intrinsic and extrinsic coagulation pathways. However, a decrease in the plasma precursor of thromboplastin (factor XI), fibrin-stabilizing factor (factor XIII) [6], and the level of the anticoagulant component-protein S [7] was revealed.

The concentration of soluble fibrin–monomer complexes (SFMC) increases, which reflects an increase in fibrin formation and correlates with an increase in fibrinogen concentration [8].

Some authors detected elevation of fibrin and fibrinogen degradation products as a result of activation of the fibrinolytic process, which reaches a maximum at 36–40 weeks [9]. The above-mentioned plateau reflects the fibrinolytic process upon activation of intravascular coagulation. Data on changes in the fibrinolytic activity of blood plasma during pregnancy are controversial: some studies show a decrease [10], others show an increase [11] or no change [12]. The fibrinolytic system, consisting of plasminogen and its activators and inhibitors, undergoes changes during normal pregnancy and in the development of complications associated with cyto-trophoblast invasion and the placental complex formation. By the end of pregnancy, the plasminogen level rises in the maternal blood circulation [12].

Data related to the reduction of fibrinolytic activity formed the basis of the hypothesis about predisposing factors to manifestation disseminated coagulation syndrome during pregnancy [13]. Such changes in indicators in hemostasis become noticeable from the beginning of the second trimester of pregnancy, while no significant changes are observed in the first trimester. However, beginning from the first weeks of gestation, in cases where the invading cytotrophoblast penetrates myometrium, mechanisms for controlling local hemostasis turn on, preventing the appearance of hemorrhages and the formation of retrochorial hematomas [14]. Decidual cells in the endometrium express tissue factor, an activator of the extrinsic coagulation pathway in the hemostasis system, which prevents bleeding from the newly formed niche when trophoblasts invade the wall of the spiral artery, as well as plasminogen activator inhibitor I (PAI-I) [13]. In addition, decidual cells are producers of tissue plasminogen activator (tPA), being an important component of the fibrinolytic system, which is involved in tissue remodeling through activation of metalloproteinases and degradation of the extracellular matrix [15]. The regulation of hemostasis processes also could be influenced by autoantibodies to plasminogen, which are detected in tiny amounts in healthy women and in high concentrations in cases of some pathologies [16]. One of the main causes of vaginal bleeding during the first trimester of pregnancy is retrochorial hematoma, which could lead to complications and chorion detachment [17]. Therefore, the identification of mothers’ indicators of impaired hemostasis, reflecting the formation of retrochorial hematomas and the possibility of developing disseminated intravascular coagulation, is important for saving the lives of the mother and child.

In managing pregnant women, the indicators of hemostasis that are mandatory for determination are prothrombin time (PT—reflects the state of the extrinsic coagulation pathway), activated partial thromboplastin time (aPTT—an indicator of the intrinsic coagulation pathway), thrombin time (TT), fibrinogen, SFMC (indicators of the common coagulation pathway) and fibrin and fibrinogen degradation products, including D-dimer, an indicator of the activity of the fibrinolytic system [18,19,20].

However, from the beginning of pregnancy, the level of D-dimer in the blood increases, reaching its peak in the third trimester and characterized by high individual variability [21]. Often, the concentration of fibrinogen degradation products in the mother’s blood does not reflect the state of the fibrinolytic system and its main component, plasminogen. In a study on the assessment of hemostatic parameters in pregnant women with retrochorial hematoma, the authors showed that elevated levels of D-dimer and hypercoagulability may indicate the formation of retrochorial hematoma [22].

We did not find any research paper devoted to the study of the activity and concentration of plasminogen in the blood of pregnant women in the first trimester with retrochorial hematomas. The effect of autoantibodies interacting with plasminogen in the course of pregnancy during this period and their role in the formation of retrochorial hematomas and chorionic detachment are also unknown. At the same time, as mentioned above, the components of the fibrinolytic system play a key role in the first months of pregnancy with intravascular invasion of the cytotrophoblast, the formation of the placenta, and the establishment of the physiological functioning of the uteroplacental region.

Thus, the aim of our study was to determine the indicators of plasminogen, as well as autoantibodies of classes A (IgA), M (IgM), and G (IgG) that bind to plasminogen, in retrochorial hematoma in the early weeks of pregnancy.

## 2. Materials and Methods

We conducted a retrospective study of 73 blood plasma samples from pregnant women (aged 18 to 40 years, gestational age 5–13 weeks), provided by the Department of Obstetrics of the Rostov State Medical University. The study group included 16 women with chorionic hematoma and chorionic detachment according to ultrasound and 57 women with a normal single pregnancy. Informed consent was obtained from all subjects involved in the study.

Criteria for exclusion from both groups included the following: acute and chronic inflammatory diseases, severe extragenital pathology, organ transplantation history, history of oncologic diseases, autoimmune disorders, (including antiphospholipid syndrome), diabetes mellitus, fetal congenital malformations, pregnancy after the use of assisted reproductive technologies, and multi-pregnancy.

Criteria for inclusion in the hematoma group were detection of chorionic hematoma and chorionic detachment from 5 to 13 weeks of gestation by means of ultrasound data (including Doppler study).

This research was approved by the Bioethics Committee at the Research Institute of Human Morphology (Protocol No. 19, 3 October 2017). The study conformed to the ethical principles for medical research involving human subjects, including research on identifiable human material and data (according to the principles of the World Medical Association and the Declaration of Helsinki).

Plasma samples were characterized in the laboratory of the Perinatal Center, Rostov-on-Don, Russia, on an Sysmex CA-1500 automatic analyzer (SYSMEX CORPORATION, Kobe, Japan) according to the main indicators of the hemostasis system: prothrombin time (PT) (“Thromborel S”, Siemens (10 × 10 mL) OUHP49), thrombin time (TT) (“Test Thrombin”, Siemens (10 × 5 mL + 1 × 50 mL) OWHM13), activated partial thromboplastin time (aPTT) (“Pathromtin SL”, Siemens (10 × 5 mL) OQGS29), fibrinogen concentration (Fibrinogen) (“Multifibren U”, Siemens (10 × 5 mL) 10446691), concentration of D-dimer (“INNOVANCE D-Dimer”, Siemens (3 × 4 ml + 2 × 3 × 5 mL + 3 × 2.6 mL + 2 × 1 mL) OPBP03), and concentration of soluble fibrin-monomer complex (SFMC) (#PG12/2, SPC Renam).

The plasminogen activity (%Plg) in blood plasma samples was determined using the Reagent kit for determining the plasminogen activity by the optical method “Reachrom-plasminogen” (SPC Renam). The plasminogen concentration was revealed by quantitative enzyme-linked immunosorbent assay according to the “sandwich” principle on 96-well polystyrene plates (Maxibinding, SPL Life Sciences Co. Ltd., Seoul, Korea) based on specific mouse monoclonal antibodies (mAbs) that may interact with the heavy chain of plasminogen. Plasminogen bound to specific monoclonal antibodies (clone Pg11, LLC Hytest, Moscow, Russia) immobilized at the bottom of a well of a polystyrene plate was detected with biotinylated mAbs against plasminogen (clone Pg7, LLC Hytest, Moscow, Russia) and streptavidin conjugate with horseradish peroxidase (LLC IMTEK, Moscow, Russia). The enzymatic reaction was shown using a solution of tetrabenzidine containing hydrogen peroxide. The enzymatic reaction was stopped with 1M sulfuric acid. Human plasminogen (#SRP6518, Sigma-Aldrich Chemical co., MO, USA) was used to prepare standards. Plasma samples calibrated with the Reachrom-Plasminogen Reagent Kit for determining the activity of plasminogen were used as control materials. The sensitivity of the system was 20 ng/mL plasminogen. Blood plasma samples were diluted with 0.5% bovine serum albumin (BSA; MP Biomedicals, LLC, Illkirch, France) in phosphate buffer solution (PBS) (0.001 M sodium phosphate with 0.1 M NaCl, pH 7.4) 200-fold.

The reference intervals for PT, aPTT, TT, Fibrinogen, SFMC, D-Dimer, and %PG in the first trimester of local pregnant women are presented in Table 1.

The levels of IgG, IgM, and IgA interacting with plasminogen were determined in blood plasma with sodium citrate by enzyme immunoassay on 96-well polystyrene plates (SPL Maxibinding, Korea), with human plasminogen immobilized at the bottom of the wells, as described previously [16]. Immunoglobulins bound to plasminogen were revealed by conjugates of specific mAbs against human IgG or IgM or IgA with horseradish peroxidase (Diateh-Em, Moscow, Russia). The enzymatic reaction was shown using a solution of tetrabenzidine containing hydrogen peroxide. The enzymatic reaction was stopped with 1M sulfuric acid. Blood plasma samples were diluted 100-fold with 0.5% BSA in PBS before ELISA. The content of IgG, IgM and IgA was expressed in conventional units (c.u.), which was estimated as the optical density value multiplied by 100 (sample dilution). %Plg/IgM-Plg was calculated as the ratio %Plg to IgM-Plg.

Statistical analysis of the obtained data was performed using the Statistica (Ver.12, StatSoft Inc., Tulsa, OK, USA) and AtteStat for Excel (Ver. 12.0.5, Gaidyshev, Orenburg, Russia) software packages. The pattern of the distribution of hemostasis parameters was determined using the Shapiro–Wilk W-test (Appendix A).

The distribution of most indicators significantly differed from the normal one; in this regard, the median and quartiles were calculated, and non-parametric comparison methods were used. Significance of differences between groups was determined using the Mann–Whitney U-test. The relationship between the indicators was determined using the Spearman correlation coefficient. The clinical significance of the indicators was assessed by ROC analysis. Differences were considered statistically significant at *p* < 0.05.

## 3. Results

Indicators of the hemostasis system in patients with normal pregnancy and those with retrochorial hematoma are presented in Table 2. The values of PT, aPTT, TT, fibrinogen, SFMC, D-dimer, and activity and concentration of PG for healthy pregnant women were within the normal range (Table 1) [21,22,23].

When determining the significance of differences between the parameters in the comparison groups (Table 1), significant differences were found in PT (*p* < 0.05) (Figure 1A), SFMC (*p* < 0.05) (Figure 1B), plasminogen activity (*p* < 0.05) (Figure 1C) and the ratio of plasminogen activity to the content of IgM -PG (*p* < 0.05) (Figure 1D).

The relationships between the parameters of the hemostasis system were estimated using the Spearman rank correlation coefficient (Table 3, Appendix A). This statistical method found direct correlations between the parameters of extrinsic (PT) and intrinsic (aPTT) (r = 0.519; *p* < 0.05), as well as intrinsic (aPTT) and general (TT r = 0.336; *p* < 0.05, fibrinogen r = 0.349; *p* < 0.05) clotting pathways in the normal pregnancy group. In the group with chorionic detachment, this relationship was not observed. In both groups, the fibrinogen concentration showed a significant direct correlation with SFMC (healthy r = 0.455; *p* < 0.005, retrochorial hematoma r = 0.732; *p* < 0.005). SFMC also correlated with D-dimer in healthy pregnant women (r = 0.436; *p* < 0.05), but not in women with retrochorial hematoma.

In normal pregnancy, in contrast to pregnancy with retrochorial hematoma, a direct correlation was observed between Plg activity (% Plg) and fibrinogen (r = 0.324, *p* < 0.05) (Table 3, Figure 2(A1,A2)). In both groups, there were no significant relationships between plasminogen activity and its concentration in the blood. In the group with retrochorial hematoma, there was a negative correlation that was higher between the indicator of the extrinsic coagulation pathway (PT) and plasminogen activity (healthy r = −0.371; *p* < 0.05, retrochorial hematoma r = −0.539; *p* < 0.05).

The study revealed a direct relationship between the intrinsic coagulation pathway index (aPTT) and the level of class M autoantibodies to plasminogen (IgM-Plg) in both groups. The correlation between aPTT and IgM-Plg in the group with chorionic detachment was significantly higher (retrochorial hematoma r = 0.510; *p* < 0.05) (Table 3, Figure 2(B2)) than in the uncomplicated pregnancy group (healthy r = 0.288; *p* < 0.05) (Figure 2(B1)). In normal pregnancy, a correlation was also found between the ratio of Plg activity (% Plg) to IgM-Plg (r = 0.408; *p* < 0.05) (Table 3). There was no correlation between these parameters in retrochorial hematoma. The level of IgM-Plg showed a direct correlation in retrochorial hematoma with TT (r = 0.545; *p* < 0.05) (Figure 2(C2)) and an inverse correlation between TT and % Plg/IgM-Plg ratio (r = −0.645; *p* < 0.05) (Figure 2(D2)). Normally, such relationships were absent (Figure 2(C1,D1)). Negative correlation between % Plg/IgM-Plg ratio and aPTT increased in women with retrochorial hematoma compared with healthy women (healthy r = −0.405; *p* < 0.05, retrochorial hematoma r = −0.613; *p* < 0.05) (Table 3). We did not find an effect of IgG-Plg and IgA-Plg on hemostasis parameters in normal pregnancy and retrochorial hematoma.

During the first trimester of pregnancy, the concentration of fibrinogen increased (r = 0.352; *p* < 0.05) (Table 3) indicating augmentation of procoagulation processes. At the same time, an increase in plasminogen activity was found (r = 0.329; *p* < 0.05) (Table 3).

To determine the diagnostic significance of % Plg and the % Plg/IgM-Plg ratio in case of chorion detachment, ROC analysis was performed (Figure 3). The test was shown to determine chorionic detachment by evaluation of plasminogen activity: sensitivity was 100%, specificity was 68.4% and AUC was 0.85. The %Plg/IgM-Plg ratio test demonstrated a sensitivity of 75%, a specificity of 94%, and AUC of 0.82.

## 4. Discussion

The influence of gestational age on changes in hemostasis parameters during normal pregnancy was consistent with data in the literature [13,23]. During the first trimester of pregnancy, the concentration of fibrinogen increased (Table 3), indicating augmentation of procoagulation processes. At the same time, plasminogen activity increased as well (Table 3). According to other research data, the fibrinolytic activity of blood in pregnant women is controversial [10,11,12]. The increase in plasminogen activity compensates for enlargement of procoagulation processes. This trend persists throughout the entire period of pregnancy [12]. With regard to retrochorial hematoma, no correlation was found between gestational age, % Plg and fibrinogen that indicated disturbances of the hemostasis system. The results of the study confirmed the data on enlargement of procoagulation mechanisms in pregnant women [13]. Moreover, they demonstrated an increase in the activity of the main component of the fibrinolytic system, plasminogen. High plasminogen levels showed the involvement of compensatory mechanisms starting from the first trimester of pregnancy, although fibrinolytic blood activity diminished in the later weeks of gestation [10,13].

In the group with retrochorial hematoma, plasminogen activity (%Plg) decreased significantly (Table 2, Figure 1C), indicating an abnormality of fibrinolysis that may be explained by the active consumption of plasminogen by the forming hematoma, or by the influence of other factors such as PAI-1, alpha-2-antiplasmin, etc. At the same time, there was no significant change in fibrinogen (Table 2), but there was a significant increase in SFMC (Table 2, Figure 1B). The fibrinogen concentration showed a significant direct correlation with SFMC in both groups (Table 3), which is not surprising, since the fibrin monomer is a product of clot formation [24]. However, the indicator of correlation was higher in the group with retrochorial hematoma (Table 3). There was a significant increase in prothrombin time in the group with hematoma (Table 2, Figure 1A), indicating a lack of clotting factors of the extrinsic coagulation pathway, which could be also actively consumed. Significant disorders in the group with retrochorial hematoma may be indicated by a change in the correlations between hemostasis indicators compared with healthy pregnant women (Table 3). Changing the direction of correlation between extrinsic coagulation pathway (PT) and plasminogen activity (%Plg) (Table 3) and taking into account a significant increase in PT (Table 2, Figure 1A) and a decrease in % Plg (Table 2, Figure 1C) may indicate disturbances of hemostasis processes from both coagulation and fibrinolytic mechanisms, indicating possible abnormal coagulation and fibrinolysis.

Evaluating the relationship between the indicator of the intrinsic coagulation pathway, aPTT, and the level of IgM-PG may indicate the influence of these immunoglobulins on the intrinsic coagulation pathway. A prolonged aPTT is associated with antiphospholipid antibodies [13]. Previously, researchers have shown that plasmin may be an important autoantigen that interacts with antiphospholipid antibodies IgG and IgM, particularly against β2 glycoprotein I [25,26]. Antibodies binding to plasmin may inhibit fibrinolytic activity or cross-react with homologous proteins in the coagulation cascade to impair hemostasis and fibrinolysis [27,28]. We were not able to assume that those IgM-Plg belong to the group of antiphospholipid antibodies because the test for antiphospholipid antibodies was not conducted in this study. All participants included in the study had no autoimmune disorders, including antiphospholipid syndrome. However, we could assume that IgM-Plg is involved in the formation of retrochorial hematomas and possible chorionic detachment. On the other hand, there was no significant difference between the level of class M autoantibodies to plasminogen in the comparison groups. Perhaps this is a disadvantage of a small sample size.

The clinical significance of plasminogen activity and the ratio of plasminogen activity to IgM-PG in the blood plasma of women in the first trimester of pregnancy, as indicators of chorionic detachment, appears to be quite high (AUC >0.8). Therefore, plasminogen levels are recommended to be taken into account as predictors of hemostasis disorders when managing pregnant women in gynecological consultations. The study of the indicator of plasminogen activity in the blood at all stages of retrochorial hematoma formation may be useful for reducing reproductive losses.

In this study, we would like to draw attention to the principal fibrinolytic factor of hemostasis during the first trimester of pregnancy and to the change in the indicator of plasminogen activity in the general circulation in association with the formation of retrochorial hematoma. The main focus of the study was on the participation of autoantibodies in the regulation of plasminogen activity, as well as the issue of identifying correlations between functional indicators of coagulation and fibrinolysis in the first trimester of gestation in healthy women and patients with retrochorial hematoma and chorionic detachment. The main criterion included in the study group was the presence of retrochorial hematoma, leading to detachment of the chorion in clinically healthy women of 5–13 gestation weeks. As a result of the exclusion of patients with gynecological and obstetrics pathology in anamnesis such as chronic pathologies of various etiologies, we evaluated only a small group of patients (n = 16).

The small sample size was the main limiting factor in this study that could influence the identification of statistical significance of differences in indicators between groups and low correlation values between indicators within groups. However, some scores showed significant differences. Intergroup differences in the indicator of the level of IgM that binds to plasminogen did not reach statistical significance, which may be due to the small sample size in the group with retrochorial hematoma.

Our study was conducted among clinically healthy women, so their hemostasis parameters were within the normal range and, despite significant differences between groups, remained within the normal range and after detachment of the chorion. Future studies need to increase sample size and consider parameters that affect plasminogen activity, such as the levels of antiphospholipid antibodies, PAI-1, PAI-2, urokinase, alpha-2-antiplasmin, a2-macroglobulin, etc.

Thus, as a result of this study, it was found that the activity of plasminogen in blood plasma samples from women with chorionic detachment was significantly lower than that in plasma samples from women in the normal pregnancy group.

For the first time, a significant direct correlation was shown between aPTT and the level of autologous IgM interacting with plasminogen, which may indicate the involvement of IgM in the regulation of the intrinsic pathway of blood coagulation. According to the results of the study, it is possible to recommend the determination of plasminogen activity in the management of pregnant women in gynecological practice.

## Figures and Tables

**Figure 1 biomedicines-10-02284-f001:**
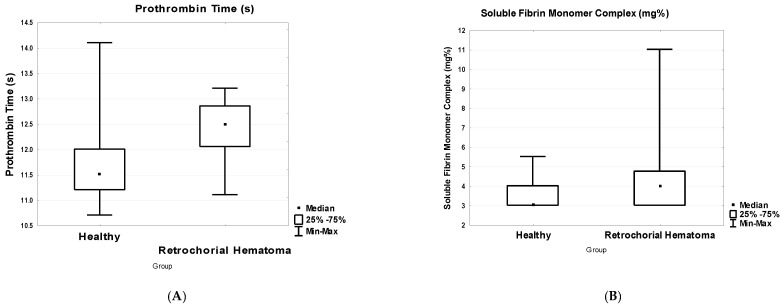
Comparisons of prothrombin time (**A**), soluble fibrin monomer complex (**B**), plasminogen activity (**C**) and ratio of plasminogen activity to IgM-Pg (**D**) in plasma of healthy pregnant women (Healthy, n = 57) and women with retrochorial hematoma (Retrochorial Hematoma, n = 16).

**Figure 2 biomedicines-10-02284-f002:**
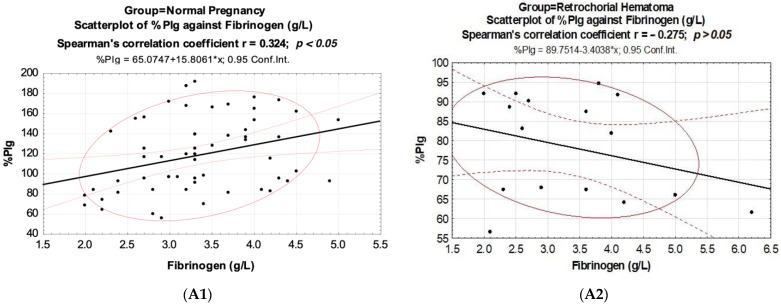
Scatterplots of hemostatic parameters in normal pregnancy (n = 56) and retrochorial hematoma (n = 16). %Plg against Fibrinogen (**A1**,**A2**), IgM-Plg against aPTT (**B1**,**B2**), TT against IgM-Plg (**C1**,**C2**) and TT against %Plg/IgM-Plg ratio (**D1**,**D2**).

**Figure 3 biomedicines-10-02284-f003:**
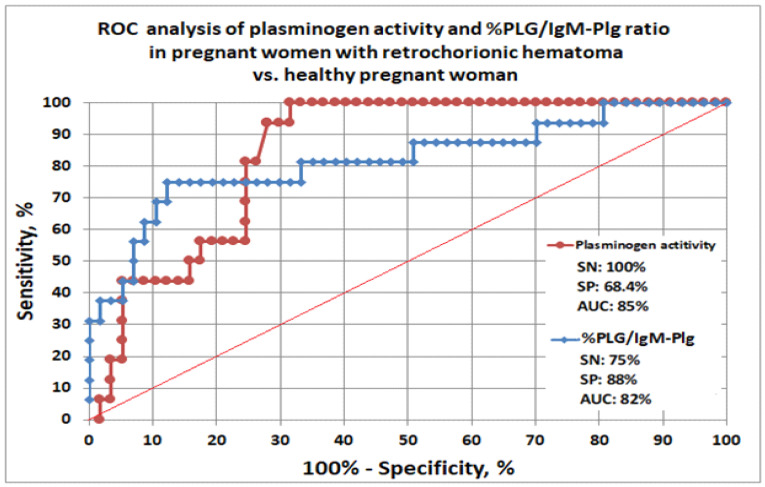
ROC analysis of plasminogen activity and %PLG/IgM-Plg ratio in pregnant women with retrochorionic hematoma (n = 16) vs. healthy pregnant women (n = 56).

**Table 1 biomedicines-10-02284-t001:** 95% reference intervals for coagulation and anticoagulation tests of first trimester pregnant women.

Index	Reference Intervals (2.5–97.5%) for Healthy Pregnant Women in First Trimester
Prothrombin time (PT), s	11–16
Activated partial thrombin time (aPTT), s	20–40
Thrombin time (TT), s	14–21
Fibrinogen, g/L	1.8–4.7
Soluble fibrin monomeric complexes (SFMC), mg%	0–5.5
D-dimer, ng/mL	10–550
Plasminogen activity (%Plg), %	80–135

**Table 2 biomedicines-10-02284-t002:** Indicators of hemostasis in patients with normal pregnancy and those with retrochorial hematoma.

Index	Normal Pregnancy, n = 57, Median (Q1;Q3)	Retrochorial Hematoma, n = 16, Median (Q1;Q3)	Significance Level of Differences according to the Mann–Whitney U Test, *p*-Value
**Prothrombin time (PT), s**	11.5 (11.2;12.0)	12.5 (12.1;12.9)	**0.0006**
Activated partial thrombin time (aPTT), s	30.6 (29.1;33.3)	32.8 (30.6;36.0)	0.0686
Thrombin time (TT), s	19.0 (18.7;19.6)	19.0 (18.1;20.8)	0.7638
Fibrinogen, g/l	3.3 (2.8;3.9)	3.3 (2.5;4.1)	0.6936
**Soluble fibrin monomeric complexes (SFMC), mg%**	3.0 (3.0;4.0)	4.0 (3.0;4.8)	**0.0082**
D-dimer, ng/ml	372 (192;704)	310 (193;931)	0.9893
**Plasminogen (PG) activity, %**	116.5 (91.8;143.4)	82.5 (66.7;91.0)	**0.00003**
Plasminogen concentration, mcg/ml	88.0 (72.0;112.1)	101.4 (80.7;129.0)	0.1802
IgG-PG, c.u.	139.5 (94.4;239.1)	126.5 (113.3;170.2)	0.6647
IgM-PG, c.u.	89.4 (67.2;119.6)	100.6 (76.5;140.3)	0.3205
IgA-PG, c.u.	141.5 (76.7;186.4)	124.2 (89.1;147.3)	0.2598
**PG/IgM-PG activity**	1.15 (1.00;1.47)	0.76 (0.53;0.97)	**0.0001**

**Table 3 biomedicines-10-02284-t003:** The degree of correlation between hemostasis parameters in the group with normal pregnancy (n = 57) and in the group with retrochorial hematoma (n = 16), Spearman’s correlation coefficient (r).

Pair of Variables	Spearman Rank Order Correlations							
	Marked Correlations Are Significant at *p* < 0.05000							
	Normal Pregnancy				Retrochorial Hematoma			
	Valid, n	Spearman, r	t(n − 2)	*p*-Value	Valid, n	Spearman, r	t(n − 2)	*p*-Value
Gestation period (weeks) & Fibrinogen (g/L)	57	0.351545	2.78489	0.007330	16	−0.100603	−0.37834	0.710851
Gestation period (weeks) & %Plg	57	0.329186	2.58541	0.012407	16	−0.165436	−0.62766	0.540333
%Plg & PT (s)	57	−0.371411	−2.96667	0.004448	16	−0.538747	−2.39274	0.031301
%Plg & Fibrinogen (g/L)	57	0.324137	2.54106	0.013902	16	−0.275405	−1.07192	0.301885
%Plg & IgM-Plg	57	0.407901	3.31324	0.001635	16	0.125920	0.47493	0.642160
%Plg & %Plg/IgM-Plg ratio	57	0.407571	3.31002	0.001651	16	0.292862	1.14604	0.270991
IgM-Plg & aPTT (s)	57	0.287595	2.227	0.030064	16	0.509588	2.21602	0.043765
IgM-Plg & TT (s)	57	−0.056935	−0.42292	0.673999	16	0.545389	2.43462	0.028885
%Plg/IgM-Plg ratio & PT (s)	57	−0.360592	−2.86711	0.005861	16	−0.442480	−1.84617	0.086115
%Plg/IgM-Plg ratio & aPTT (s)	57	−0.404720	−3.28232	0.001792	16	−0.613119	−2.90393	0.011552
%Plg/IgM-Plg ratio & TT (s)	57	−0.131466	−0.98351	0.329663	16	−0.644546	−3.15430	0.007031
%Plg/IgM-Plg ratio & Fibrinogen (g/L)	57	0.383476	3.07935	0.003235	16	−0.133922	−0.50565	0.620973
%Plg/IgM-Plg ratio & IgM-Plg	57	−0.605662	−5.64482	0.000001	16	−0.863871	−6.41696	0.000016
Fibrinogen (g/L) & PT (s)	57	−0.385628	−3.09964	0.003052	16	0.760149	4.37742	0.000632
Fibrinogen (g/L) & aPTT (s)	57	−0.349048	−2.76235	0.007788	16	0.341446	1.35926	0.195557
Fibrinogen (g/L) & TT (s)	57	−0.279248	−2.15676	0.035412	16	−0.039114	−0.14647	0.885642
Fibrinogen (g/L) & SFMC (mg%)	57	0.455123	3.79062	0.000375	16	0.732157	4.02194	0.001261
SFMC (mg%) & Plg, ug/ml	57	−0.030017	−0.22271	0.824586	16	0.534996	2.36937	0.032731
SFMC (mg%) & D-dimer (ng/mL)	57	0.436431	3.59734	0.000689	16	0.009145	0.03422	0.973185
D-dimer (ng/mL) & IgA-Plg	57	−0.342696	−2.70532	0.009067	16	−0.266372	−1.03403	0.318653
PT (s) & aPTT (s)	57	0.519143	4.50465	0.000035	16	0.365115	1.46744	0.164360
aPTT (s) & TT (s)	57	0.336078	2.64635	0.010590	16	0.396896	1.61794	0.127977

## Data Availability

The data presented in this study are available on request from the corresponding author.

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
