# Peer review of "Hemostasis System and Plasminogen Activity in Retrochorial Hematoma in the First Trimester of Pregnancy"

_biomedicines, 2022, doi:10.3390/biomedicines10092284_

Round 1
Reviewer 1 Report
The manuscript by Tikhonova et al reported that in a cohort of 16 women with retrochorial hematoma resulting in chorionic detachment and 57 women with normal pregnancy, plasminogen (PG) activity was significantly lowered among those with retrochorial hematoma, while prothrombin time (PT) and soluble fibrin monomeric complexes (SFMC) were increased with p-value lower than 0.05. Concentrations of plasminogen and antibodies against plasminogen (IgG-PG, IgM-PG, IgA-PG) did not statistically significant changes, PG activity/IgM-PG is indeed lower (most likely due to lowered PG activity). ROC curve plotted using PG activity & PG activity/IgM-PG ratio to classify retrochorionic hematoma vs normal pregnancy achieved AUC of 0.85 & 0.82 respectively, suggesting potential of monitoring PG activity for management/prevention of retrochorial hematoma during pregnancy.
The study investigate an important issue especially since there are limited information on changes in fibrinolytic pathway and it’s relationships to pregnancy complications & miscarriages. However, there are several issues limiting the current study. Can the authors please consider the following carefully and revise their manuscript accordingly?
1. the cohort is small. Granted activity of plasminogen appeared to reach statistical significance, I have reservations that with larger cohort, lab-to-lab variations etc statistical significance in other parameters such as PT & SFMC may not be retained (a difference of 1s in PT is really small). Can the authors please comment on the limitation of cohort size (and any other limitations of the study) in the last paragraph of their discussion? Similarly, the correlation data presented in Table 2 are generally low although some reaches statistical significance, again hampered by low sample numbers.
2. What is the diagnostic criteria used for retrochorial hematoma? And there is no results showing that the outcome of all 16 subjects with retrochorial hematoma is miscarriage? Retrochorial hematoma and chorionic detachment appeared to be considered as equal in the manuscript, but the readers were not presented with any data that this is indeed the case in the study.
3. Please use ‘intrinsic’ vs ‘extrinsic’ coagulation pathways, instead of ‘internal’ vs ‘external’. Terms such as ‘precursor of thromboplastin’ may be used previously but we now know it is not the right description of factor XI. I think factor XI, factor XIII etc are now well-established in the literature in the last few decades that may be used as is.
4. Line 48: what do the authors mean by “which plateaus the most pronounced plateau”?
5. Line 67: PAI-1 is “plasminogen activator inhibitor 1”
6. Line 77: is “disseminated coagulation syndrome” the right term? Or is it “disseminated intravascular coagulation”?
7. Line 169: again, “fibrinolyticopathy” does not seem like a commonly used term?
8. Line 170: the drop in plasminogen activity may or may not be due to consumption of plasminogen, since the assay essentially measure the activity of plasmin after addition of streptokinase. Increase in plasma concentrations of PAI-1, alpha-2-antiplasmin etc may also affect the activity. I think there is a missed opportunity hear that the authors did not check for the above inhibitors, and should again be discussed as a limitation to the study. In any case, the sentence should be revised to reflect that the decrease in PG activity is not exclusively due to consumption.
9. Line 179: “retrochorial”
10. Line 213: “prolonged” is probably a better description that “elongation”
Author Response
Dear Reviewer,
We are most grateful to you for critical remarks, correcting and valuable advice.
We have tried to give you comprehensive answers to the remarks. In the new manuscript version, we have corrected most of the typos and inaccuracies you mentioned. Please see the attachment.

Reviewer 2 Report
Dear authors,
thank you very much for presenting your work for publication.
The topic is off interest and the question about the course of plasminogen during pregnancy is relevant.
However, I have some severe concerns. What about the ethical considerations? Did you request an ethical board review, if so provide, please. Did you inform the patients and get an informed consent?
What about power-analysis? Did you estimate the number of patients you would need?
Did you collect demographic data of the patients? What about information on co-morbidities and medication? Did you check and exclude patients with abnormal coagulation?
Why did you combine the results and the discussion?
There is no good reflection and comparison to the literature.
Please find more comments in the pdf-file.
Greetings

Author Response

(The authors gave the same response as above.)

Round 2
Reviewer 1 Report
The authors have answer most of the points raised in the previous round of review except the following:
1. the cohort is small. Granted activity of plasminogen appeared to reach statistical significance, I have reservations that with larger cohort, lab-to-lab variations etc statistical significance in other parameters such as PT & SFMC may not be retained (a difference of 1s in PT is really small). Can the authors please comment on the limitation of cohort size (and any other limitations of the study) in the last paragraph of their discussion? Similarly, the correlation data presented in Table 2 are generally low although some reaches statistical significance, again hampered by low sample numbers.
While I appreciate that recruitment of participants was challenging, what I am asking for is some comments from the authors on the limitations of this study in terms of cohort size and the effect the small cohort size has on the conclusions drawn. Please include a paragraph outlining these limitations in the discussion.
8. Line 170: the drop in plasminogen activity may or may not be due to consumption of plasminogen, since the assay essentially measure the activity of plasmin after addition of streptokinase. Increase in plasma concentrations of PAI-1, alpha-2-antiplasmin etc may also affect the activity. I think there is a missed opportunity hear that the authors did not check for the above inhibitors, and should again be discussed as a limitation to the study. In any case, the sentence should be revised to reflect that the decrease in PG activity is not exclusively due to consumption.
I do not see this point being included in the authors’ reply at all.
Author Response
Dear Reviewer,
Thank you for your comments on our article.
We have added several suggestions in accordance with your comments at the conclusion of the new version.
Please review the application.

Reviewer 2 Report
Dear authors,
thank you for revising your manuscript.
Adopting most of the suggestions improved the quality clearly.
I added only a few minor comments to be addressed in the pdf-file.

Author Response

(The authors gave the same response as above.)
